# Application of the Coupled Markov Chain in Soil Liquefaction Potential Evaluation

**Hsiu-Chen Wen** **, An-Jui Li \*, Chih-Wei Lu and Chee-Nan Chen**

Department of Civil and Construction Engineering, National Taiwan University of Science and Technology, Taipei 106, Taiwan
\* Correspondence: laj871@mail.ntust.edu.tw

**Abstract:** The evaluation of localized soil-liquefaction potential is based primarily on the individual evaluation of the liquefaction potential in each borehole, followed by calculating the liquefaction-potential index between boreholes through Kriging interpolation, and then plotting the liquefaction-potential map. However, misjudgments in design, construction, and operation may occur due to the complexity and uncertainty of actual geologic structures. In this study, the coupled Markov chain (CMC) method was used to create and analyze stratigraphic profiles and to grid the stratum between each borehole so that the stratum consisted of several virtual boreholes. The soil-layer parameters were established using homogenous and random field models, and the subsequent liquefaction-potential-evaluation results were compared with those derived using the Kriging method. The findings revealed that within the drilling data range in this study, the accuracy of the CMC model in generating stratigraphic profiles was greater than that of the Kriging method. Additionally, if the CMC method incorporated with random field parameters were to be used in engineering practice, we recommend that after calculating the curve of the mean, the COV should be set to 0.25 as a conservative estimation of the liquefaction-potential interval that considers the evaluation results of the Kriging method.

**Keywords:** coupled Markov chain; soil profile; random field; liquefaction potential

## 1. Introduction

Located on the Pacific Ring of Fire, Taiwan's geology is characterized by fragile geologic features, a high fault density, frequent earthquakes, and soil liquefaction. Currently, the hyperbolic function (HBF) method is the most widely adopted approach for measuring the soil-liquefaction potential in Taiwan [1]. The steps involved in this method include assessing the liquefaction potential of various boreholes, calculating the liquefaction-potential index between the boreholes through Kriging interpolation, and, finally, generating a map showing the liquefaction potential of different areas. While different liquefaction-potential-assessment methods are used in other parts of the world, the Kriging method remains the most widely used method for generating liquefaction-potential-risk maps [2–7].

A challenge involved with these types of assessments is that the geologic structures in reality are complex and riddled with uncertainty [8,9]. Due to the practical limitations of geological survey techniques and project budgets, only a limited number of boreholes can be drilled in a project [10]. The Taiwanese government faces similar constraints in its current implementation of extensive surveys on the liquefaction potential in various cities across the country. To this end, the geological data can only be acquired accurately onsite at the exact borehole location, as it is difficult to acquire data in other locations [11]. Consequently, this may lead to misjudgments in designs, construction, and operations. To address this challenge, this study integrates the methods for assessing soil-liquefaction potential and geological uncertainty in order to acquire more realistic geological data.

There are two types of geological uncertainty [12]: (1) the spatial variability of soil properties in homogeneous soil layers [13,14], and (2) the geological uncertainty in heterogeneous soil layers [15,16]. Analyzing the spatial variability of soil properties in geotechnical engineering has generated much attention over the past few decades [17–23]. However, the geological uncertainty in heterogeneous soil layers significantly affects the geological structure and performance [12,24,25]. The methods for modeling the geological uncertainty in heterogeneous soil layers can be divided into two categories. The first category relies on variance-function modeling in geostatistics [26], such as Kriging [27], Gaussian thresholding [28], and multiple-point statistics [10]. However, this method is strongly dependent on the quality of specific locations and projects, as well as the availability of sufficient borehole data. The second method utilizes Markov models, which include Markov random field [28,29] and Markov chain [30] models. Previous studies have proposed the estimation of geological uncertainty by using geological models based on the Markov random field theory [25,31,32]. This approach has been used to assess the impact of geological uncertainty on slope stability [10,33] and tunnels [11]. However, this approach requires preexisting knowledge of the strata direction [34,35]. Elfeki and Dekking proposed the coupled Markov chain (CMC) model to simulate the geological uncertainty in a heterogeneous soil layer with an unknown strata direction [15]. Many studies have improved the CMC model and applied new models to resolve various geotechnical engineering problems, such as geological uncertainty [16,34–37], slope evaluation [38], tunnel construction [39], and soil seepage [40]. More recently, several researchers have proposed other approaches to simulating geological uncertainty, such as nonparametric methods [41] and random-field methods [42,43]. However, compared to other methods, the CMC method requires fewer parameters, has high applicability, and is clear, reliable, and easy to interpret [44,45]. Thus, this study applied the CMC method to simulate the geological uncertainty in heterogeneous soil layers.

The four most common methods for assessing soil-liquefaction potential are the Taiwan HBF method [1], the U.S. National Center for Construction Education and Research (NCEER) method [46], the Architectural Institute of Japan (AIJ) method [47], and the Japan Road Association (JRA) method [48,49]. The HBF and NCEER methods were developed based on the simplified procedure proposed by the American professor, H.B. Seed. In particular, the HBF method is distinguished by the number of case studies it includes; in addition to more than 300 worldwide datasets [50], it also contains 300 datasets specific to the 1999 Chi-Chi earthquake, making it the only method with a large number of actual soil-liquefaction cases in Taiwan. Another method commonly used in Taiwan is the JRA method, developed in 1996. It has been validated using datasets from six earthquakes, 64 liquefaction cases, and 23 non-liquefaction cases [51], and was recently revised in 2017 [49]. Therefore, this study adopts the HBF and JRA methods to evaluate soil-liquefaction potential.

In recent years, a growing number of studies have adopted random-field models for evaluating soil-liquefaction potential [42,52–55]. However, most of these studies focus on the influence of the spatial variability of soil properties on liquefaction-potential evaluation. Therefore, this study first applied the CMC method to create and analyze stratigraphic profiles and to grid the stratum between each borehole to ensure that the stratum contains several virtual boreholes. Subsequently, the HBF and JRA methods were used to evaluate the liquefaction potential at each borehole, in which the stratum parameters were determined according to the homogenous and random field methods. Finally, the analysis results were compared to those acquired through the Kriging method (involving interpolation of the liquefaction-potential index between boreholes) to generate liquefaction-potential-evaluation results that more accurately represented realistic conditions.

## 2. Materials and Methods

### 2.1. Coupled Markov Chain (CMC)

The Markov chain is a random model proposed by Russian mathematician Andrey Markov in 1909. It describes the random process within a state space by which one state

is converted to another. The process must be memoryless, as the random distribution of the subsequent state can only be determined by the current state and is unassociated with events that occur before the current state. In Equation (1), when $Z_0, Z_1, Z_2, \ldots, Z_m$ form a random variable sequence obtained from the state space $(S_1, S_2, \ldots, S_n)$, the sequence is known as a Markov chain or Markov process:

$$
\begin{aligned}
\Pr(Z_i = S_k | Z_{i-1} = S_l, Z_{i-2} = S_n, Z_{i-3} = S_r, \ldots, Z_0 = S_p) \\
= \Pr(Z_i = S_k | Z_{i-1} = S_l) =: p_{lk}
\end{aligned}
\tag{1}
$$

where | is a conditional probability.

### 2.1.1. Transition Probability Matrix

In a one-dimensional problem, a Markov chain can be described by a single transition-probability matrix. A transition probability is the relative frequency with which one state converts to another. In Equation (2), these transition probabilities can be arranged into a matrix form, as follows:

$$
p = \begin{bmatrix}
p_{11} & p_{12} & \cdot & \cdot & p_{1n} \\
p_{21} & \cdot & \cdot & \cdot & \cdot \\
\cdot & \cdot & p_{lk} & \cdot & \cdot \\
\cdot & \cdot & \cdot & \cdot & \cdot \\
p_{n1} & \cdot & \cdot & \cdot & p_{nn}
\end{bmatrix}
\tag{2}
$$

where $p_{lk}$ represents the probability of state $S_l$ converting to state $S_k$, and $n$ is the number of states in the system. Thus, the probability of state $S_l$ converting to states $S_1, S_2, \ldots, S_n$ is indicated in $p_{1l}$ in the first row, and $l = 1, 2, \ldots n$, and so on. The transition probability matrix $p$ must have the following features: (a) its elements cannot be negative, and (b) the sum of each row of elements is 1.

### 2.1.2. Markov Chain

Krumbein (1968) first used a one-dimensional Markov model to construct stratigraphic sequences [56]. A one-dimensional Markov chain event is shown in Figure 1. Assuming that the state of a preceding unit cell $i - 1$ is $S_l$, and the state of the unit cell $N$ is $S_q$, then the probability that the state of the unit cell $i$ is $S_k$ can be represented mathematically as follows:

$$
\Pr(Z_i = S_k | Z_{i-1} = S_l, Z_N = S_q)
\tag{3}
$$

Through derivations, Equation (3) transforms to Equation (4):

$$
p_{lk|q} = \frac{p_{lk} p_{kq}^{(N-i)}}{p_{lq}^{(N-i+1)}}
\tag{4}
$$

in which $p_{lk|q}$ is the probability that the state of the unit cell $i$ is $S_k$, the state of a preceding unit cell $i - 1$ is $S_l$, and the state of the unit cell $N$ is $S_q$.

In Equation (4), when the unit cell $i$ is separated away from unit cell $N$, $p_{lq}^{(N-i+1)}$ and $p_{kq}^{(N-i)}$ ineffective, because they are seemingly equivalent to a stationary probability. However, when we approach unit cell $N$, its state becomes effective, and the simulation results are influenced by the state of unit cell $N$.

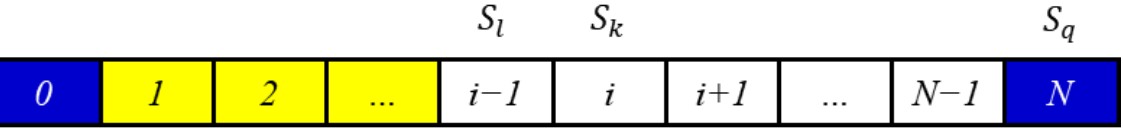

**Figure 1.** Event-sequence numbering in a one-dimensional Markov chain.

### 2.1.3. Two-Dimensional Markov Chain

Because the concepts supporting the means to define "the future" in a two-dimensional system remain unclear, it is challenging to extend the future state conditions of a one-dimensional Markov chain to a two-dimensional Markov chain. Amro Elfeki and Michel Dekking, in 2001, proposed an extremely simple yet low-cost approximation method [15] in which the one-dimensional transition probabilities at the vertical and horizontal directions are placed into a CMC, as shown in Figure 2. This method also assumes that if the state of a preceding unit cell $i-1, j$ is $S_l$, and the state of the unit cells $i, j-1$ and $N_x, j$ is $S_m$ and $S_q$, respectively, then the probability that the state of the unit cell $i, j$ is $S_k$ can be represented mathematically in Equation (5), as follows:

$$\Pr\left(Z_{i,j} = S_k \big| Z_{i-1,j} = S_l, Z_{i,j-1} = S_m, Z_{N_x,j} = S_q\right) \tag{5}$$

Through derivations, Equation (5) transforms to Equation (6):

$$
\begin{aligned}
p_{lm,k|q} &:= \Pr\left(Z_{i,j} = S_k \big| Z_{i-1,j} = S_l, Z_{i,j-1} = S_m, Z_{Nx,j} = S_q\right) \\
&= \frac{p_{lk}^h \cdot p_{kq}^{h(N_x-i)} \cdot p_{mk}^v}{\sum_f p_{lf}^h \cdot p_{fq}^{h(N_x-i)} \cdot p_{mf}^v} k = 1, \ldots n
\end{aligned}
\tag{6}
$$

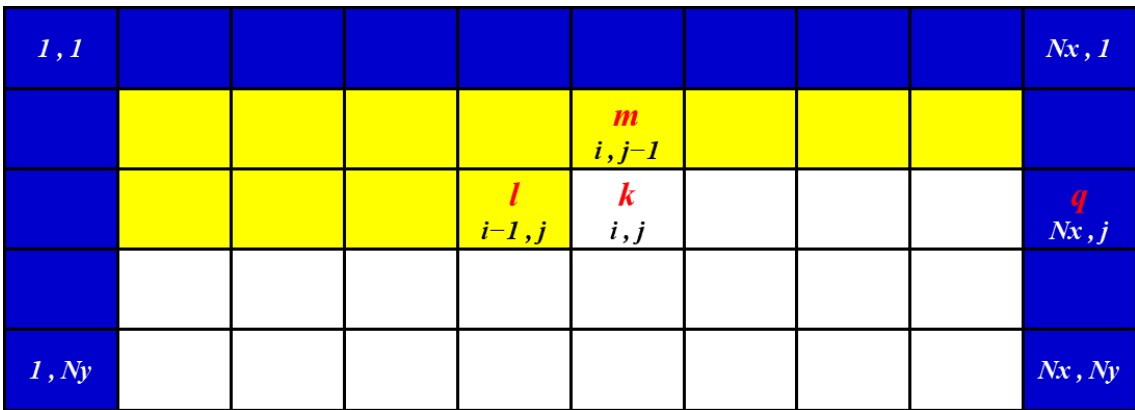

**Figure 2.** Two-dimensional field-numbering system in a coupled Markov chain.

### 2.1.4. Estimation of the Transition-Probability Matrix

The vertical-transition-probability matrix (VTPM) and horizontal-transition-probability matrix (HTPM) are two important input parameters for a CMC model.

VTPM($p^v$) can be directly estimated based on the borehole data [16]. First, the geologic profile is divided into numerous units of the same size, each with a soil type. Next, each element ($p_{lk}^v$) in VTCM($p^v$) can be obtained through Equation (7):

$$p_{lk}^v = \frac{T_{lk}^v}{\sum_{f=1}^n T_{lf}^v} \tag{7}$$

where $T_{lk}^v$ represents the number of vertical observations in which state $S_l$ converts to state $S_k$.

However, unlike VTCM ($p^v$), the elements in HTPM ($p^h$) cannot be directly obtained using the borehole data, because the boreholes are not continuous on the horizontal direction, in addition to the long interval between them. Thus, this study applied the calculation process and method proposed by Cao, W. et al. [34], Walther's law, and the drilling data to evaluate HTPM ($p^h$), as shown in Equation (8). According to Walther's law, in lithology,

the depositional sequences are similar on the vertical and horizontal directions but have different magnitudes [16,44,57].

$$T_{lk}^h = \left\{ \begin{array}{ll} T_{lk}^v & l \neq k \\ K T_{lk}^v & l = k \end{array} \right. \tag{8}$$

Thus, it is evident that the Walther's constant ($K$) is the only parameter required to calculate HTPM ($p^h$). The value of $K$ can be estimated based on the actual borehole data, and using an approach similar to the maximum-likelihood-estimation method [16,58].

### 2.2. Liquefaction-Potential Evaluation

Iwasaki et al. proposed the liquefaction-potential index (LPI) to determine the soil-liquefaction potential [51]. The factor of safety (FS) of each soil layer without a surface structure needs to be considered when calculating LPI. Regarding the FS calculations, a simplified method is usually employed in engineering practice. In the simplified method, FS can be defined as cyclic-resistance ratio (CRR) divided by cyclic-stress ratio (CSR). CSR and CRR are seen as liquefaction loading and liquefaction resistance, respectively. CSR and CRR are calculated through empirical formulas. In this study, the HBF and JRA methods were adopted to evaluate the FS of each soil layer. The main input parameters are related to earthquake and soil information. The earthquake parameters have peak ground acceleration and the earthquake magnitude. In general, a desired condition should be assumed, such as a real earthquake event or designed earthquakes. The soil parameters include standard penetration test N-value (SPT-N), effective stress, and fines content. They can be estimated by field drilling and laboratory testing.

#### 2.2.1. Hyperbolic Function (HBF) [1]

This method is primarily based on the base framework of the simplified procedure developed by Seed et al. [59], in which the degree of the liquefaction resistance of soil is determined based on the data of in situ liquefaction and non-liquefaction areas during an earthquake. The HBF method includes more than 300 datasets of worldwide cases [50], as well as 300 datasets pertaining to the 1999 Chi-Chi earthquake. During regression analysis, the liquefaction resistance of soil is represented by a hyperbolic function and, thus, the HBF method is considered as a liquefaction-evaluation method developed based on the Chi-Chi earthquake data. The analytical process of the HBF method is shown in Figure 3.

#### 2.2.2. JRA Method (2017) [48,49]

The JRA method, in which soil-liquefaction tests were performed indoors on a large number of high-quality onsite samples, which were vibrated for 20 h, was developed by Iwasaki et al. (1982) [60] and Tatsuoka et al. (1980) [61]. Liquefaction potential is evaluated based on the relationship between the derived liquefaction resistance SR20 and the SPT-N value. The reliability of the JRA method has been validated through6 earthquakes, 64 liquefaction cases, and 23 non-liquefaction cases [51], and was recently revised in 2017 [49].

Following the Great Hanshin earthquake in 1995, the JRA revised its 1990 method due to the remarkably higher liquefaction values and the strength of the earthquake. In particular, it reassessed the types of soil required for liquefaction evaluation, underestimated the liquefaction behaviors of soils with high N values, considered the influence of fines on the liquefaction-resistance strength, and included the fine content (FC, in %) in the evaluation. The analytical procedure is shown in Figure 4. The characteristics of this method are as follows: (a) the basis for comparison is the maximum cyclic-stress ratio, instead of the mean cyclic-stress ratio; (b) the design of earthquake parameters only requires the peak ground acceleration (PGA), and not the magnitude of an earthquake (M).

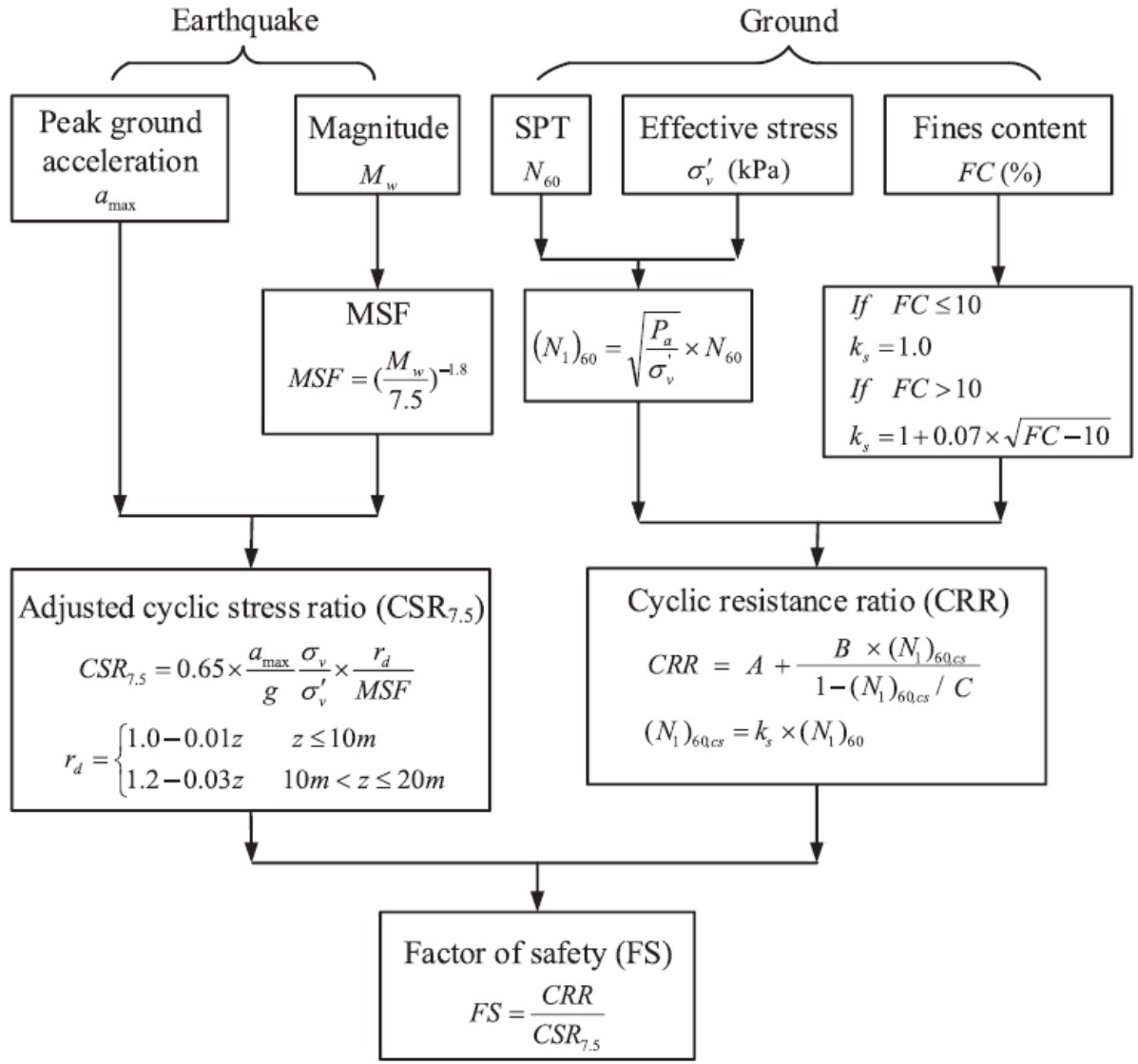

**Figure 3.** Analytical procedure of the HBF method [1].

*2.3. Drilling Data*

This study utilized the drilling data of a construction project in northern Taiwan, as shown in Figure 5, and analyzed its A-A′ profile. The main texture classes were clay, silty sand and silt in this project. The A-A′ profile was categorized using the Unified Soil Classification System (USCS), and consisted of CL, ML, SM, and SP-SM, as shown in Figure 6. The mean number and coefficient of variation (COV) of each soil Liquefaction potential-related parameter is presented in Table 1. Additionally, when evaluating the liquefaction potential, the groundwater level (GWL) and the maximum ground acceleration ($A_{max}$) of the examined region during an earthquake must be set. Based on the drilling data of the project, the GWL was $-5.5$ m underground, and the designed earthquake acceleration of the region was 0.32 g.

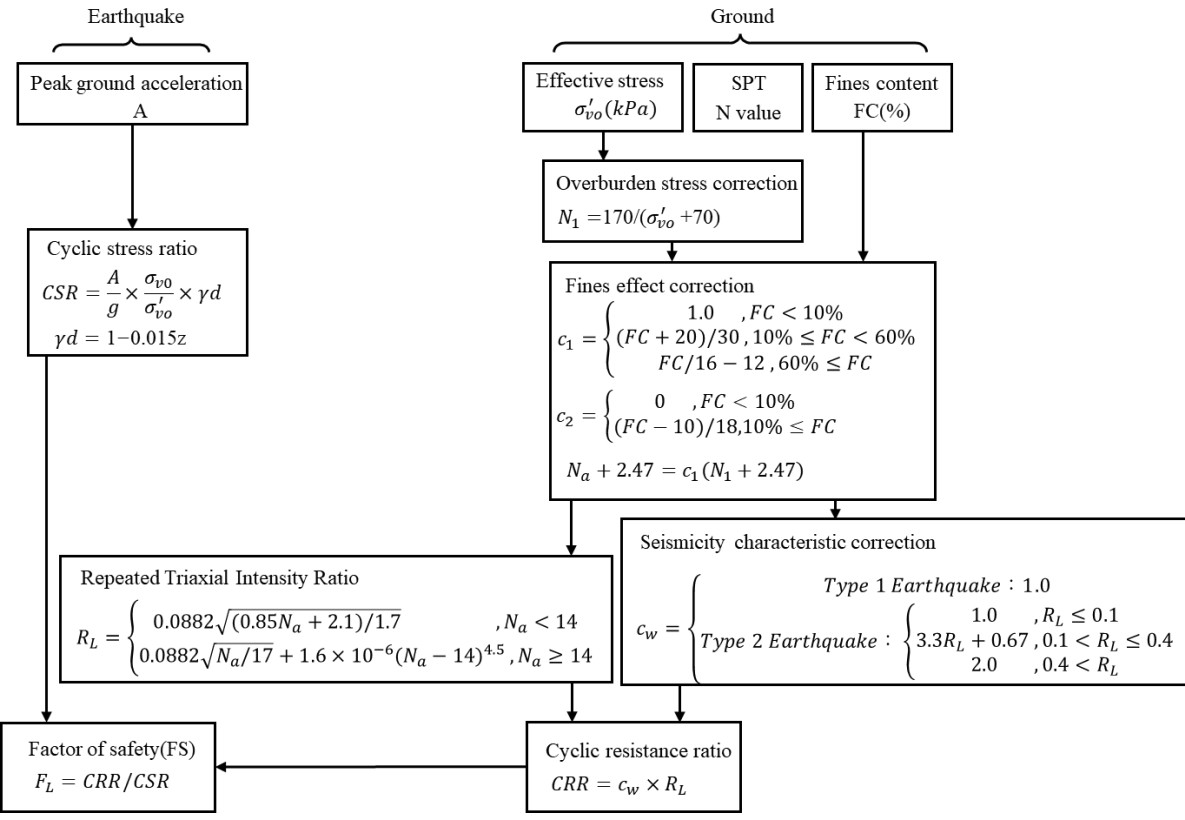

**Figure 4.** Analytical procedure of the JRA (2017) method [48,49].

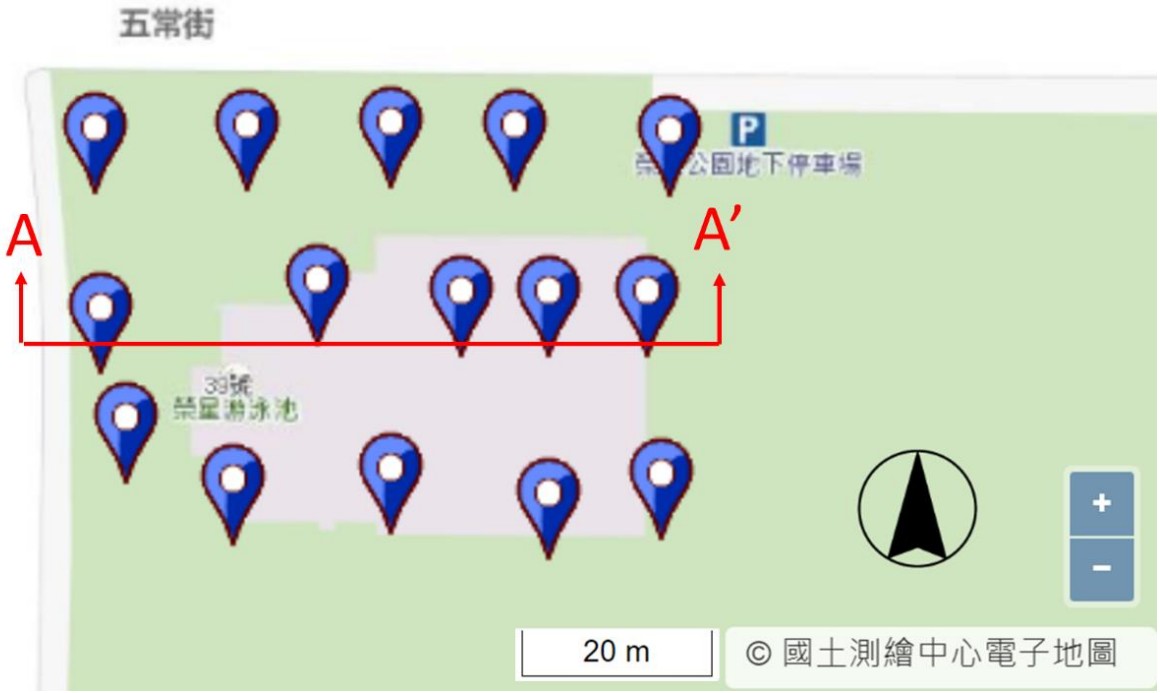

**Figure 5.** Schematic of the drilling locations at a construction site in northern Taiwan.

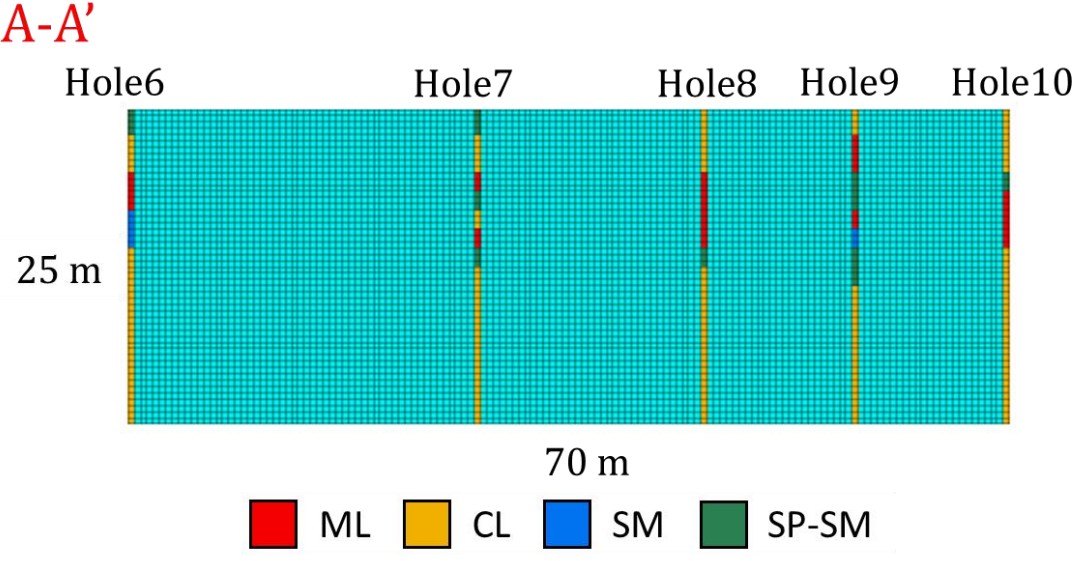

**Figure 6.** Soil-distribution profile of actual boreholes along A-A′.

**Table 1.** The relevant parameters and coefficients of variance in the liquefaction-potential evaluation of a construction project in northern Taiwan.

| USCS | $\gamma_t$ Mean (kN/m³) | $\gamma_t$ COV | N Mean | N COV | FC Mean (%) | FC COV |
|------|------|------|------|------|------|------|
| ML | 18.64 | 0.04 | 7 | 0.67 | 70 | 0.21 |
| CL | 18.15 | 0.04 | 6 | 0.43 | 97 | 0.05 |
| SM | 19.03 | 0.05 | 9 | 0.40 | 45 | 0.45 |
| SP-SM | 18.25 | 0.06 | 10 | 0.21 | 10 | 0.74 |

## 3. Results and Discussion

### 3.1. Testing the Accuracy of the CMC and Selecting the Suitable Grid Size

To evaluate the accuracy of the simulated stratum, we set Holes 7, 8, and 9 in Figure 6 as observation holes (which were not subjected to stratigraphic analysis). The scheme is shown in Table 2, and the borehole-accuracy index [35] is defined as:

$$I_d = \frac{1}{N_r} \sum_{i=1}^{N_r} \frac{G_{d,i}}{N_z} \times 100\% \tag{9}$$

where:

$N_r$ is the total number of realizations by using the enhanced Markov chain model.

$G_{d,i}$ is the number of cells whose soil types are consistent with that in observation borehole $d$.

$N_z$ is the number of rows in borehole $d$.

**Table 2.** Different borehole schemes considered in the study.

| Borehole Scheme | Hole 6 | Hole 7 | Hole 8 | Hole 9 | Hole 10 |
|------|------|------|------|------|------|
| Scheme 1 | ✓ | | ✓ | ✓ | ✓ |
| Scheme 2 | ✓ | ✓ | | ✓ | ✓ |
| Scheme 3 | ✓ | ✓ | ✓ | | ✓ |

To compare the stratigraphic profiles generated through the Kriging and CMC methods, we first used both methods to generate the stratigraphic profiles and gridding the data for the three schemes in Table 2, and the results as shown in Figures 7 and 8, respectively.

The stratigraphic profile generated through the CMC method was based on the procedure and method proposed by Cao et al. [34], in which the VTPM, *K* and HTPM of each scheme was derived, as shown in Tables 3–5. Lastly, we used Equation (9) to deduce the accuracy, as shown in Table 6. Based on the drilling data in this study, the CMC method has a 70% accuracy in stratigraphic-profile generation, which is higher than that observed using the Kriging method. This means that the CMC method can better build stratigraphic profiles.

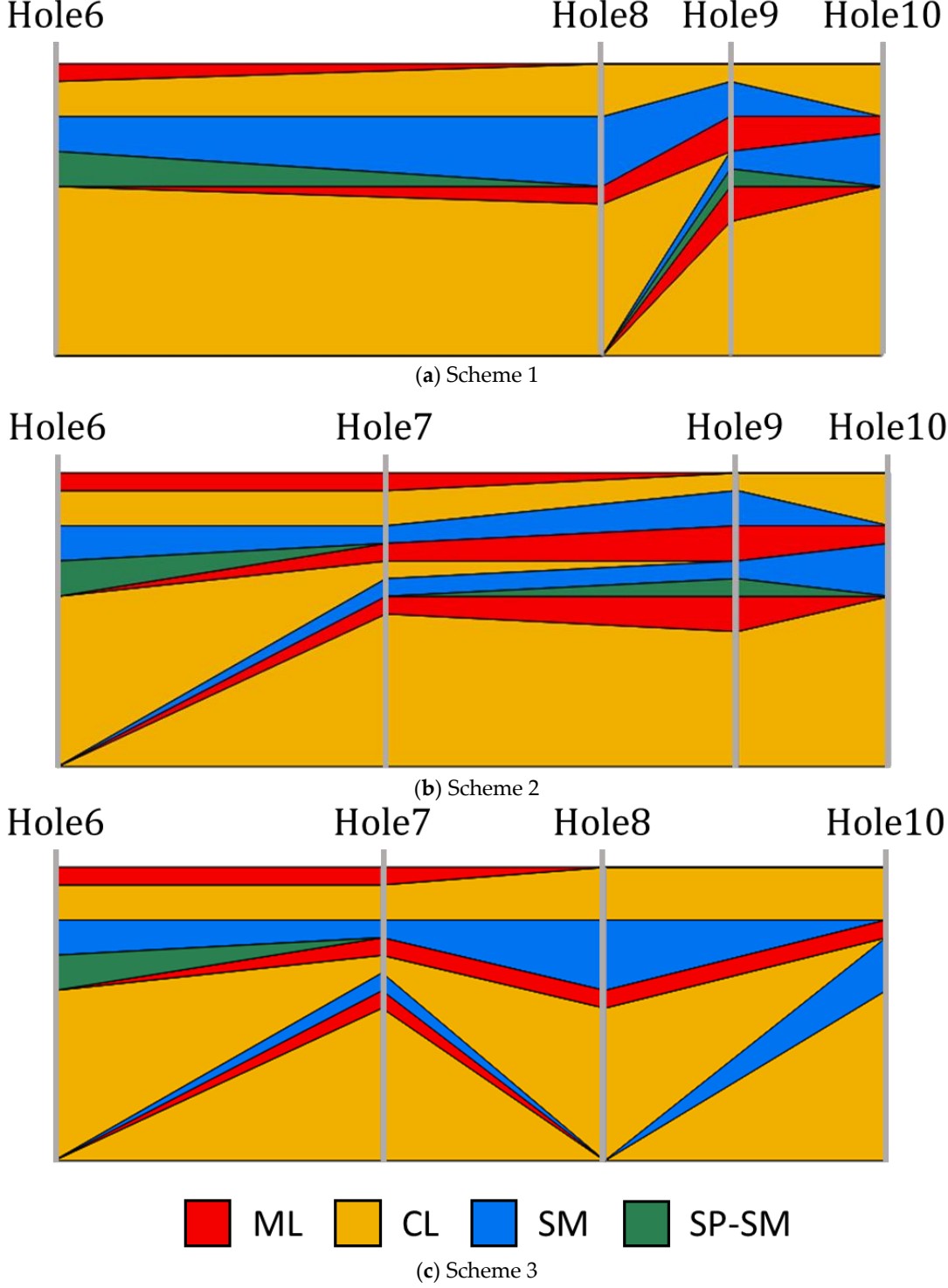

(**a**) Scheme 1

(**b**) Scheme 2

(**c**) Scheme 3

**Figure 7.** Stratigraphic profiles generated through Kriging interpolation.

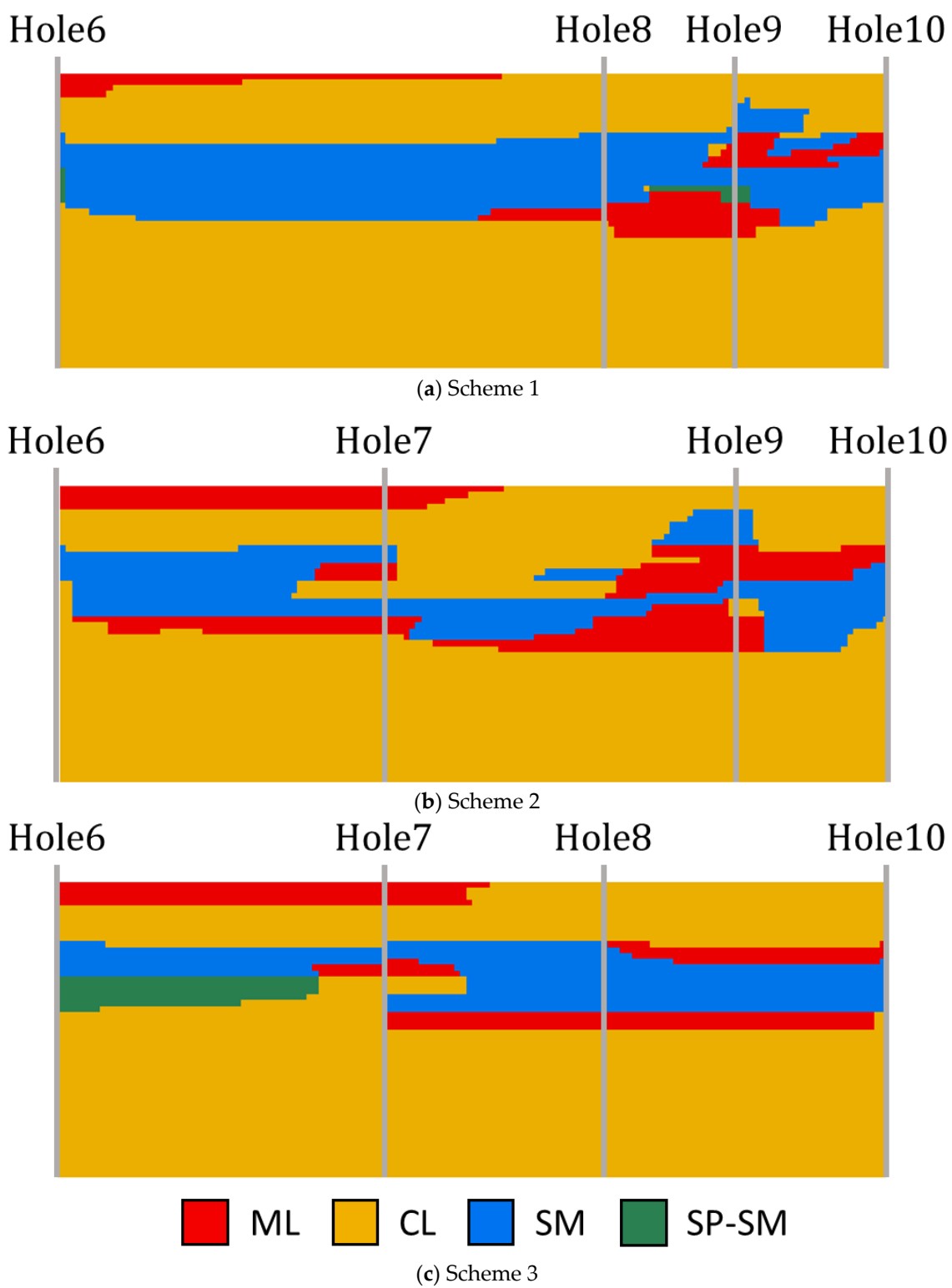

Figure 8. Stratigraphic profiles generated through the CMC method.

**Table 3.** The estimated VTPMs for various borehole schemes.

| (a) Scheme 1 | | | | |
|---|---|---|---|---|
| **Soil Type** | **ML** | **CL** | **SM** | **SP-SM** |
| **ML** | 0.7727 | 0.1364 | 0.0909 | 0.0000 |
| **CL** | 0.0077 | 0.969 | 0.0233 | 0.0000 |
| **SM** | 0.0555 | 0.0278 | 0.8611 | 0.0556 |
| **SP-SM** | 0.1111 | 0.1111 | 0.0000 | 0.7778 |
| (b) Scheme 2 | | | | |
| **Soil Type** | **ML** | **CL** | **SM** | **SP-SM** |
| **ML** | 0.7586 | 0.1724 | 0.0690 | 0.0000 |
| **CL** | 0.0078 | 0.9609 | 0.0313 | 0.0000 |
| **SM** | 0.1000 | 0.0333 | 0.8000 | 0.0667 |
| **SP-SM** | 0.1111 | 0.1111 | 0.0000 | 0.7778 |
| (c) Scheme 3 | | | | |
| **Soil Type** | **ML** | **CL** | **SM** | **SP-SM** |
| **ML** | 0.7000 | 0.2500 | 0.0500 | 0.0000 |
| **CL** | 0.0073 | 0.9635 | 0.0292 | 0.0000 |
| **SM** | 0.0909 | 0.0303 | 0.8485 | 0.0303 |
| **SP-SM** | 0.0000 | 0.1667 | 0.0000 | 0.8333 |

**Table 4.** The estimated values of K for various borehole schemes.

| **Borehole Scheme** | $K_{LR}$ | $K_{RL}$ | $K_{MAX}$ | **Simulation Sequence** |
|---|---|---|---|---|
| Scheme 1 | 4.1 | 6.6 | 6.6 | From right to left |
| Scheme 2 | 9.2 | 10.1 | 10.1 | From right to left |
| Scheme 3 | 22.9 | 15.1 | 22.9 | From left to right |

**Table 5.** The estimated HTPMs for various borehole schemes.

| (a) Scheme 1 | | | | |
|---|---|---|---|---|
| **Soil Type** | **ML** | **CL** | **SM** | **SP-SM** |
| **ML** | 0.9573 | 0.0256 | 0.0171 | 0.0000 |
| **CL** | 0.0012 | 0.9952 | 0.0036 | 0.0000 |
| **SM** | 0.0095 | 0.0048 | 0.9762 | 0.0095 |
| **SP-SM** | 0.0207 | 0.0207 | 0.0000 | 0.9586 |
| (b) Scheme 2 | | | | |
| **Soil Type** | **ML** | **CL** | **SM** | **SP-SM** |
| **ML** | 0.9695 | 0.0218 | 0.0087 | 0.0000 |
| **CL** | 0.0008 | 0.9960 | 0.0032 | 0.0000 |
| **SM** | 0.0121 | 0.0040 | 0.9758 | 0.0081 |
| **SP-SM** | 0.0138 | 0.0138 | 0.0000 | 0.9724 |
| (c) Scheme 3 | | | | |
| **Soil Type** | **ML** | **CL** | **SM** | **SP-SM** |
| **ML** | 0.9816 | 0.0153 | 0.0031 | 0.0000 |
| **CL** | 0.0003 | 0.9984 | 0.0013 | 0.0000 |
| **SM** | 0.0046 | 0.0015 | 0.9924 | 0.0015 |
| **SP-SM** | 0.0000 | 0.0087 | 0.0000 | 0.9913 |

**Table 6.** Comparison of the accuracy of the Kriging and CMC methods (100 cyc.).

| Borehole Scheme | Accuracy (%) | | | | |
| --- | --- | --- | --- | --- | --- |
| | Hole No. | | | | |
| | 6 | 7 | 8 | 9 | 10 |
| Scheme 1 Kriging | - | 55.3 | - | - | - |
| Scheme 1 CMC | - | 71.6 | - | - | - |
| Scheme 2 Kriging | - | - | 52.8 | - | - |
| Scheme 2 CMC | - | - | 76.3 | - | - |
| Scheme 3 Kriging | - | - | - | 44.3 | - |
| Scheme 3 CMC | - | - | - | 70.0 | - |

In the Kriging method, the liquefaction potential of each borehole was evaluated, and then the liquefaction potential index between each borehole was calculated through linear interpolation. Thus, the number of boreholes determines the quantity of the results, and the values between boreholes are represented as linear connections. It has relatively short overall calculation time for liquefaction potential evaluation. Based on the drilling data in this study, the liquefaction potential evaluation was performed only 5 times for 5 real boreholes in the Kriging method. However, In the CMC method, the soil layer between each borehole is gridding. The stratum consists of numerous virtual boreholes with a sufficient density. The liquefaction potential evaluation was performed 140 times for 5 real boreholes and 135 virtual boreholes when the grid size was set to 0.5 m. In addition, the soil layer distribution acquired in each calculation differs, the liquefaction potential evaluation results may differ slightly. An interval will be generated based on the 100 cycles performed in this study. It will cause the CMC method to be more time consuming than the Kriging method.

Given that the grid number affects the computational speed, this study examined the accuracy of the CMC model using different grid sizes (0.5 m, 1 m, 2 m, 2.5 m) for a total of 100 cycles, as shown in Table 7 and Figure 9. The results showed a significant decline in the accuracy at a grid size of 2 m. This study suggests that the grid size should be noted when performing stratigraphic profile analysis using the CMC model. Based on our drilling data, the grid size should be no larger than 1.5 m.

**Table 7.** Accuracy of the CMC model (100 cyc.) under different grid sizes.

| Scheme | Grid Size (m) | $K_{LR}$ | $K_{RL}$ | $K_{MAX}$ | Simulation Sequence | Accuracy (%) | | | | |
| --- | --- | --- | --- | --- | --- | --- | --- | --- | --- | --- |
| | | | | | | Hole No. | | | | |
| | | | | | | 6 | 7 | 8 | 9 | 10 |
| 1 | 0.5 | 4.1 | 6.6 | 6.6 | From right to left | - | 71.6 | - | - | - |
| | 1.0 | 2.0 | 3.2 | 3.2 | From right to left | - | 71.4 | - | - | - |
| | 2.0 | 1.0 | 1.6 | 1.6 | From right to left | - | 71.6 | - | - | - |
| | 2.5 | 1.0 | 1.3 | 1.3 | From right to left | - | 69.5 | - | - | - |
| 2 | 0.5 | 9.2 | 10.1 | 10.1 | From right to left | - | - | 76.3 | - | - |
| | 1.0 | 4.5 | 4.9 | 4.9 | From right to left | - | - | 75.8 | - | - |
| | 2.0 | 2.1 | 2.4 | 2.4 | From right to left | - | - | 75.2 | - | - |
| | 2.5 | 1.7 | 1.9 | 1.9 | From right to left | - | - | 73.2 | - | - |
| 3 | 0.5 | 22.9 | 15.1 | 22.9 | From left to right | - | - | - | 70.0 | - |
| | 1.0 | 11.4 | 7.5 | 11.4 | From left to right | - | - | - | 69.9 | - |
| | 2.0 | 5.6 | 3.7 | 5.6 | From left to right | - | - | - | 69.5 | - |
| | 2.5 | 4.4 | 2.9 | 4.4 | From left to right | - | - | - | 68.0 | - |

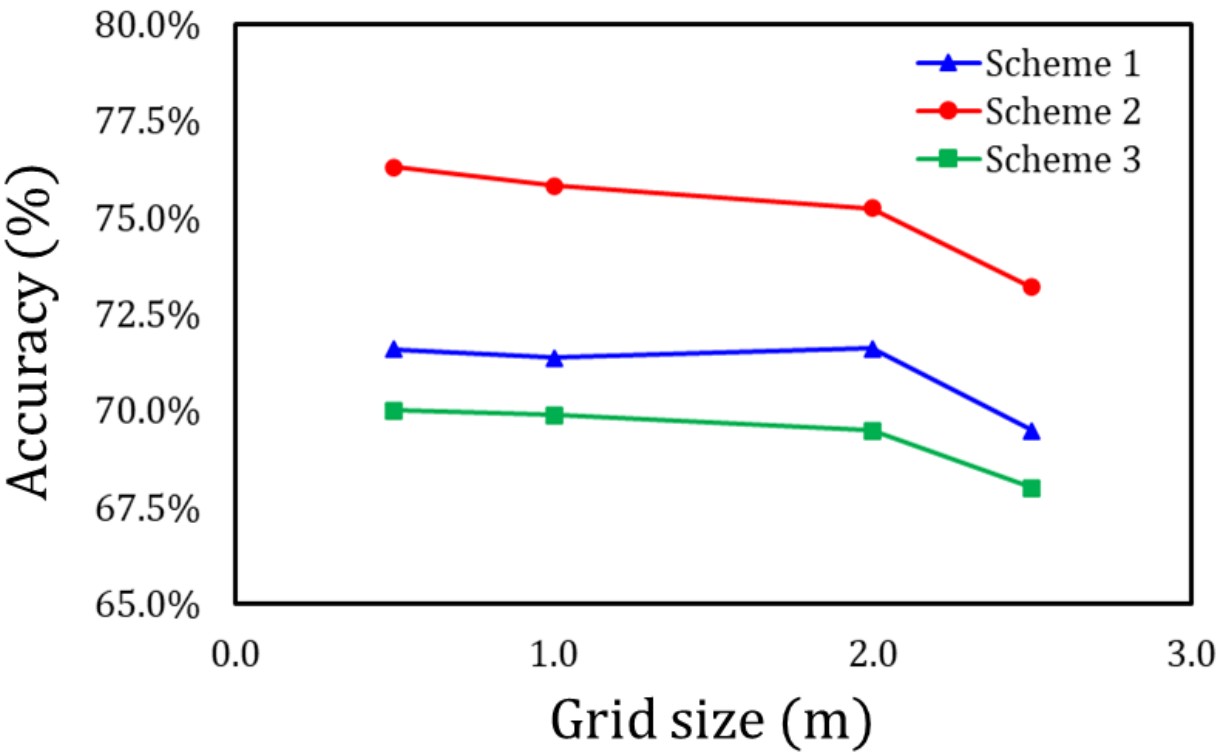

**Figure 9.** Accuracy curves of the CMC model at different grid sizes (100 cyc.).

*3.2. Comparison and Analysis of the Soil Liquefaction Potential Evaluations under the Two Methods*

This study utilized three models, including (1) the Kriging method, (2) the CMC method incorporating homogenous soil parameters, and (3) the CMC method incorporating random field-soil parameters. The overlaid liquefaction potential evaluation results for all three models derived using the HBF and JRA methods are shown in Figures 10 and 11, respectively.

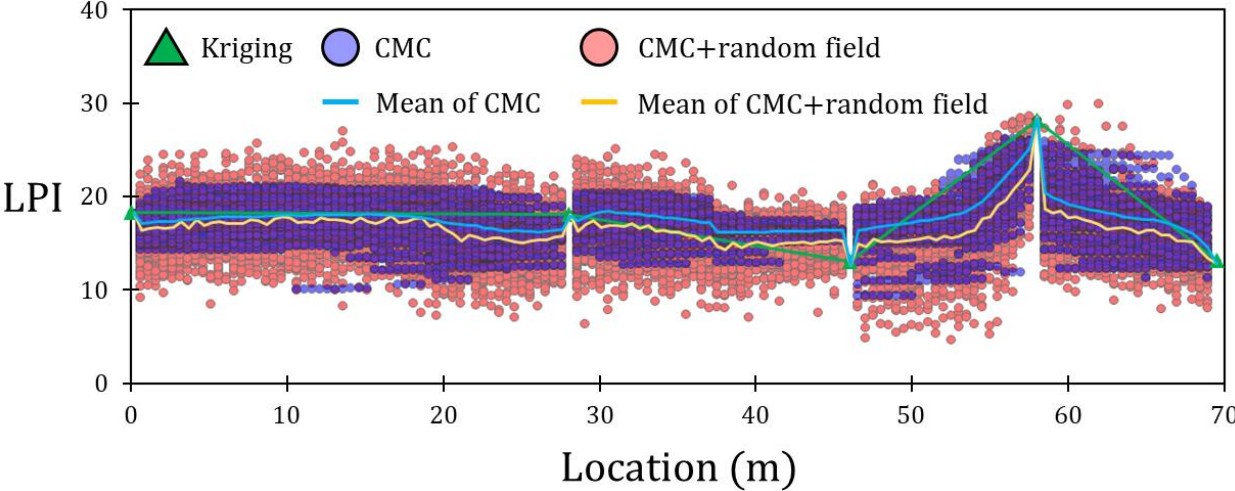

**Figure 10.** Comparison of the liquefaction-potential-evaluation results of three models through the HBF method (100 cyc.).

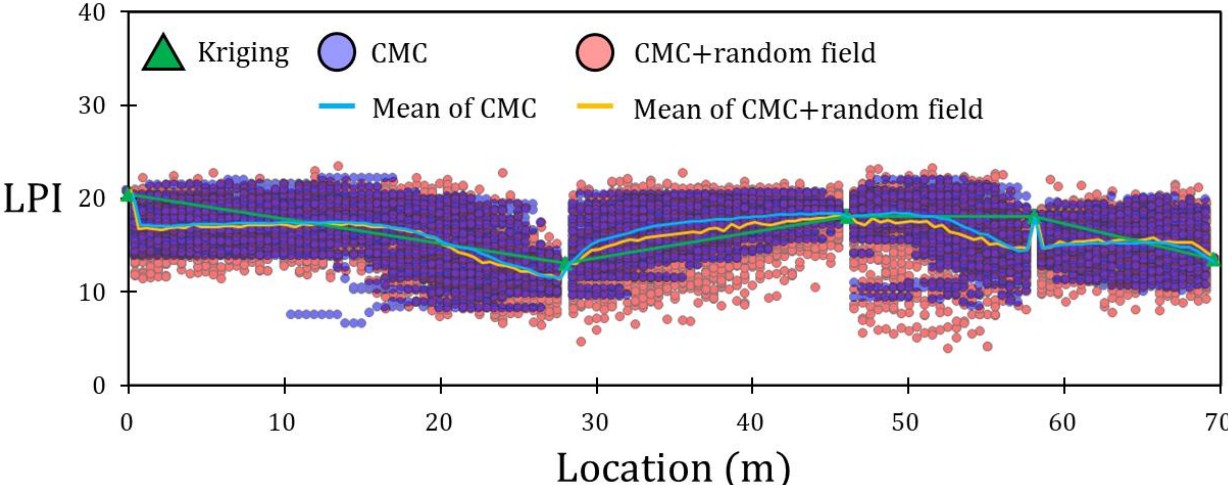

**Figure 11.** Comparison of the liquefaction-potential-evaluation results of three models through the JRA method.

Most of the results of the Kriging method (see the green lines in Figures 10 and 11) were within the limits of the results derived through the CMC method. Furthermore, in Location = 46 and 57 in Figure 10 and Location = 0 and 57 in Figure 11, the Kriging-method results are close to the extreme values in the CMC-method results. Nonetheless, this finding was acquired through actual borehole analysis. Therefore, this would be a disadvantage of the Kriging method. This means that the liquefaction potential may be overestimated or underestimated if the Kriging method only is used. When evaluating the liquefaction potential using the HBF approach, the average results of every virtual borehole generated through the CMC method incorporating homogenous-soil parameters and random-field parameters were 0.74–2.2 and 0.68–1.17 times higher than those of the Kriging method, respectively. When evaluating the liquefaction potential using the JRA approach, the average results of every virtual borehole generated through the CMC method incorporating homogenous-soil parameters and random-field parameters were 0.81–1.17 and 0.81–1.09 times higher than those of the Kriging method, respectively. This study does not discuss whether the HBF approach or the JRA approach is more suitable for Taiwan because this would require more actual liquefaction data.

In Figures 10 and 11, the results of the CMC method incorporating homogenous-soil parameters and random-field parameters are shown in light purple and pink, respectively, while the overlay region is presented in dark purple. Comparing both methods, the latter generated a larger liquefaction-potential-result interval than the former in most cases. This indicates that the effects of geological uncertainty can be accounted for more cautiously through the CMC method incorporating random-field parameters.

To help engineers utilize the CMC method incorporating the random-field-parameters model for evaluating the soil-liquefaction potential, we compared the results derived through 30 and 100 simulation cycles, as shown in Figures 12 and 13, respectively. In all three realizations, the maximum COV was 0.25, while the curves of the mean in the 30 and 100 simulation cycles were extremely similar. If the CMC method incorporating random-field parameters is to be used in engineering practice, we suggest that after calculating the curve of the mean, the COV should be set at 0.25 as a cautious estimation of the liquefaction-potential interval, which includes the evaluation results of the Kriging method.

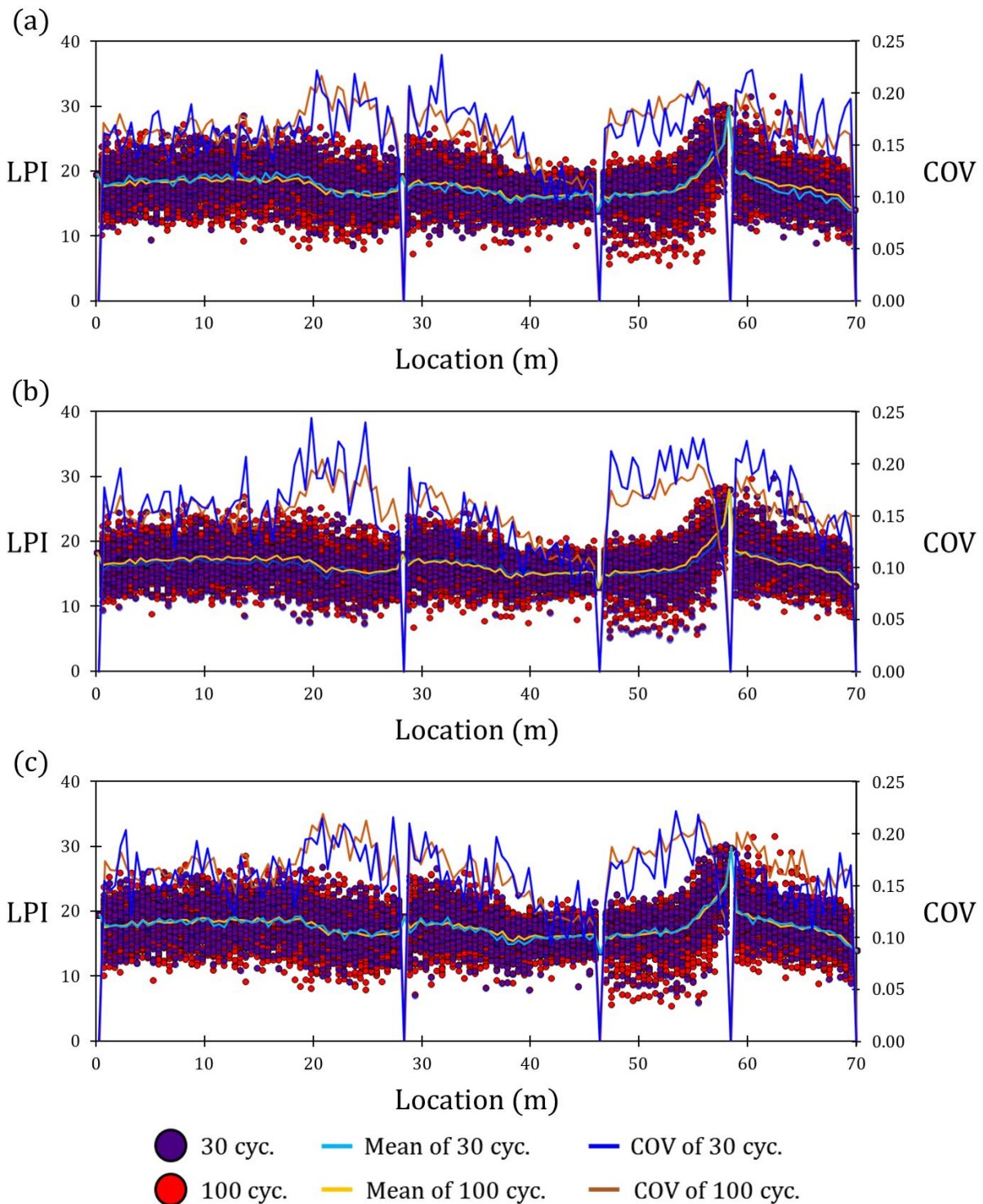

**Figure 12.** Comparison of 30 and 100 cycles of liquefaction-potential evaluations of the CMC random-field-parameter model through the HBF method. (**a**) Realization 1. (**b**) Realization 2. (**c**) Realization 3.

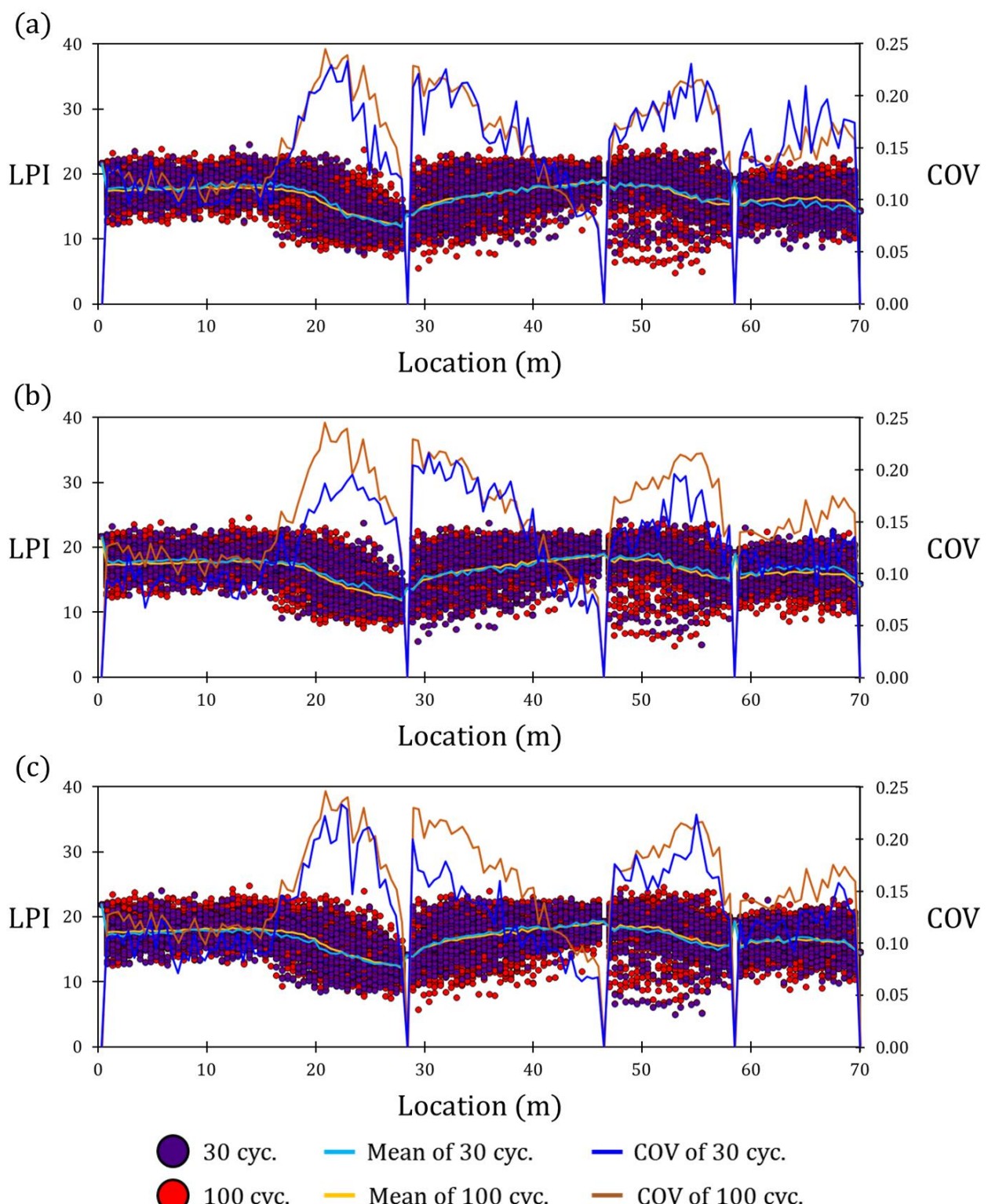

**Figure 13.** Comparison of 30 and 100 cycles of liquefaction-potential evaluations of the CMC random-field-parameter model through the JRA method. (**a**) Realization 1. (**b**) Realization 2. (**c**) Realization 3.

## 4. Conclusions

This study first generated stratigraphic profiles through the CMC method and then gridded the soil layer between each borehole so that the stratum consisted of numerous virtual boreholes. Next, we evaluated the liquefaction potential by setting the soil-layer parameters through homogenous and random-field methods. Lastly, we compared the

results generated through the traditional method (evaluating the liquefaction potential for each borehole and then calculating the liquefaction-potential index through Kriging interpolation). The following conclusions are only applicable to Taiwan, due to regional geological characteristics and drilling quality issues.

Based on the drilling data in this study, more accurate stratigraphic profiles can be created by using the CMC method than by using the Kriging method. However, if the soil layer between each borehole is gridded in the CMC method, this creates a significant computational workload for the CMC method compared to the Kriging method. Therefore, the grid size should be properly considered when performing stratigraphic-profile analysis using the CMC model. Based on our drilling data, the grid size should be smaller than 1.5 m.

Most of the Kriging-method results in our study were within the limits of the results of the CMC method. However, some of the Kriging-method results were close to the extreme values in the CMC-method results, indicating that the liquefaction potential may have been overestimated or underestimated. Therefore, it is more prudent to use the CMC method for liquefaction-potential evaluation. Based on the drilling data in this study, when evaluating liquefaction potential using the HBF and JRA approach, the average results of every virtual borehole generated through the CMC method incorporating random-field parameters were 0.68–1.17 times and 0.81–1.09 times higher than those of the Kriging method, respectively. Therefore, we suggest that after calculating the curve of the mean, the COV should be set at 0.25 as a cautious estimation of the liquefaction-potential interval when engineers use the CMC method incorporating random-field parameters for liquefaction-potential evaluation.

Comparing the CMC methods incorporating homogenous-soil parameters and random-field parameters, the latter generated a larger liquefaction-potential-result interval than the former in most cases. This indicates that the effects of geological uncertainty can be accounted for more cautiously through the CMC method incorporating random-field parameters.

**Author Contributions:** Conceptualization, H.-C.W. and A.-J.L.; Methodology, H.-C.W. and A.-J.L.; Software, H.-C.W.; Validation, A.-J.L.; Formal analysis, H.-C.W. and A.-J.L.; Investigation, H.-C.W.; Resources, C.-W.L.; Data curation, H.-C.W. and C.-N.C.; Writing—original draft, H.-C.W.; Writing—review & editing, A.-J.L. and C.-W.L.; Visualization, H.-C.W.; Supervision, A.-J.L., C.-W.L. and C.-N.C.; Project administration, C.-W.L.. All authors have read and agreed to the published version of the manuscript.

**Funding:** The authors would like to thank the Taiwan Building Technology Center from The Featured Areas Research Center Program within the framework of the Higher Education Sprout Project by the Ministry of Education in Taiwan for their support.

**Institutional Review Board Statement:** Not applicable.

**Informed Consent Statement:** Not applicable.

**Data Availability Statement:** The drilling data is a government public profile that was obtained from Engineering Geological investigation Databank of Central Geological Survey, Ministry of Economic Affairs, R.O.C. It is available at https://geotech.moeacgs.gov.tw/imoeagis/Home/Map (accessed on 15 October 2022).

**Conflicts of Interest:** The authors declare no conflict of interest.

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
