# Peer review of "Application of the Coupled Markov Chain in Soil Liquefaction Potential Evaluation"

_buildings, doi:10.3390/buildings12122095_

Round 1

Reviewer 1 Report

Dear colleagues!

I have read your manuscript with pleasure and consider it worthy of publication in Buildings. This is an interesting approach that makes it possible to simplify the assessment of liquefaction potential, which is important for the risk of geo-disasters. My non-obligatory remark concerns the possibility of improving the perception of your methodological part. I advise you to briefly describe the concept of liquefaction potential and its indicators, in particular, FS, before analyzing the assessment methods (paragraph 2.2). Also, in paragraph 2.3, it is better to decipher texture classes (silt, silty sand, etc.) at the beginning.

Best regards, Your Reviewer

Reviewer 2 Report

This is a well written paper that describes an alternate approach for interpolating liquefaction potential between boreholes. The authors are commended for a well written and well constructed research project.  The topic of liquefaction interpellation between known points is an important one for construction engineering. This paper should be useful for that purpose.

Author Response

The authors are grateful for the comment.

Reviewer 3 Report

In this article, the CMC method was used to create and analyze stratigraphic profiles and to gridding the stratum between each borehole  The soil layer parameters were established using homogenous and random field models. The liquefaction potential evaluation results were compared with those derived using the Kriging method. Recommendations were written

The present paper has practical usefulness. The article is correctly written. The description of the methods and results is clear.

Author Response

作者對評論表示感謝。